# Implementation of Practical Surface SARS-CoV-2 Surveillance in School Settings

Victor J. Cantú,[a] Pedro Belda-Ferre,[b,c] Rodolfo A. Salido,[a] Rebecca Tsai,[b] Brett Austin,[d] William Jordan,[d] Menka Asudani,[d] Amanda Walster,[d] Celestine G. Magallanes,[e,f] Holly Valentine,[e,f] Araz Manjoonian,[g,h] Carrissa Wijaya,[g] Vinton Omaleki,[g] Karenina Sanders,[b] Stefan Aigner,[c,f,i] Nathan A. Baer,[c] Maryann Betty,[b,c,j] Anelizze Castro-Martínez,[c] Willi Cheung,[c,f,h] Evelyn S. Crescini,[c] Peter De Hoff,[c,e,f] Emily Eisner,[c] Abbas Hakim,[c] Bhavika Kapadia,[c] Alma L. Lastrella,[c] Elijah S. Lawrence,[c] Toan T. Ngo,[c] Tyler Ostrander,[c] Shashank Sathe,[c,f,i] Phoebe Seaver,[c] Elizabeth W. Smoot,[c] Aaron F. Carlin,[k] Gene W. Yeo,[c,f,i] Louise C. Laurent,[e,f] Anna Liza Manlutac,[d] Rebecca Fielding-Miller,[g] Rob Knight[a,l,m]

[a]Department of Bioengineering, University of California San Diego, La Jolla, California, USA
[b]Department of Pediatrics, University of California San Diego, La Jolla, California, USA
[c]Expedited COVID Identification Environment (EXCITE) Laboratory, Department of Pediatrics, University of California San Diego, La Jolla, California, USA
[d]San Diego County Public Health Lab, San Diego, California, USA
[e]Department of Obstetrics, Gynecology, and Reproductive Sciences, University of California San Diego, La Jolla, California, USA
[f]Sanford Consortium of Regenerative Medicine, University of California San Diego, La Jolla, California, USA
[g]Herbert Wertheim School of Public Health, University of California San Diego, La Jolla, California, USA
[h]San Diego State University, San Diego, California, USA
[i]Department of Cellular and Molecular Medicine, University of California San Diego, La Jolla, California, USA
[j]Rady Children's Hospital, San Diego, California, USA
[k]Division of Infectious Diseases and Global Public Health, Department of Medicine, University of California San Diego, La Jolla, California, USA
[l]Department of Computer Science and Engineering, University of California San Diego, La Jolla, California, USA
[m]Center for Microbiome Innovation, University of California San Diego, La Jolla, California, USA

**ABSTRACT** Surface sampling for SARS-CoV-2 RNA detection has shown considerable promise to detect exposure of built environments to infected individuals shedding virus who would not otherwise be detected. Here, we compare two popular sampling media (VTM and SDS) and two popular workflows (Thermo and PerkinElmer) for implementation of a surface sampling program suitable for environmental monitoring in public schools. We find that the SDS/Thermo pipeline shows superior sensitivity and specificity, but that the VTM/PerkinElmer pipeline is still sufficient to support surface surveillance in any indoor setting with stable cohorts of occupants (e.g., schools, prisons, group homes, etc.) and may be used to leverage existing investments in infrastructure.

**IMPORTANCE** The ongoing COVID-19 pandemic has claimed the lives of over 5 million people worldwide. Due to high density occupancy of indoor spaces for prolonged periods of time, schools are often of concern for transmission, leading to widespread school closings to combat pandemic spread when cases rise. Since pediatric clinical testing is expensive and difficult from a consent perspective, we have deployed surface sampling in SASEA (Safer at School Early Alert), which allows for detection of SARS-CoV-2 from surfaces within a classroom. In this previous work, we developed a high-throughput method which requires robotic automation and specific reagents that are often not available for public health laboratories such as the San Diego County Public Health Laboratory (SDPHL). Therefore, we benchmarked our method (Thermo pipeline) against SDPHL's (PerkinElmer) more widely used method for the detection and prediction of SARS-CoV-2 exposure. While our method shows superior sensitivity (false-negative rate of 9% versus 27% for SDPHL), the SDPHL pipeline is sufficient to support surface surveillance in indoor settings. These findings are important since they show that existing investments in infrastructure can be leveraged to slow

Address correspondence to Rebecca Fielding-Miller, rfieldingmiller@health.ucsd.edu, or Rob Knight, rknight@health.ucsd.edu.
The authors declare no conflict of interest.
For a companion article on this topic, see https://doi.org/10.1128/mSystems.00109-22.

the spread of SARS-CoV-2 not in just the classroom but also in prisons, nursing homes, and other high-risk, indoor settings.

**KEYWORDS** COVID, environmental sampling, public health, SARS-CoV-2, qPCR

Over the past 2 years, the COVID-19 pandemic has claimed the lives of over 5 million people worldwide (1). Due to high density occupancy of indoor spaces for prolonged periods of time, schools are often of concern for transmission, leading to widespread school closings to combat pandemic spread when cases rise. However, K-12 schools are important resources for communities, which, besides education and childcare, often provide food, authoritative and trusted information, and a sense of belonging and security (2, 3). Therefore, alternative approaches that keep children in school are highly desirable. Performing pediatric clinical testing, such as SARS-CoV-2 detection via RT-qPCR, is expensive, difficult from a consent perspective, and increasingly politicized. Wastewater testing, although highly effective even at the level of individual cases and buildings (4), can only identify SARS-CoV-2 among the subset of individuals that defecate at school, and often cannot provide spatial resolution at finer levels. A complementary method that we have deployed in Safer At School Early Alert (SASEA) (5) is surface sampling, which allows detection of SARS-CoV-2 from surfaces within a classroom. This is a screening method, not a diagnostic; however, even without high sensitivity, surface sampling provides visibility into environments where individuals will not consent to testing and where cases are not picked up through wastewater. In particular, surface sampling can localize cases within a single classroom (5).

In previous work, we showed SARS-CoV-2 persists on a range of school-relevant surfaces using our SASEA workflow, based on collecting swabs into a 0.5% sodium dodecyl sulfate (SDS) wt/vol solution (Acros Organics, 230420025), performing nucleic acid extraction on the Kingfisher Flex liquid-handling robot (Thermo Scientific), and performing RT-qPCR using the QuantStudio 7 (5). We implemented this protocol in the research phase of SASEA (3, 6). However, this protocol requires specialized reagents and equipment that is not generally available to public health laboratories, and we needed to test its generality using workflows already operational in the San Diego County Public Health Laboratory (SDPHL), which employs Viral Transport Medium (VTM) (NEST Scientific USA, 202016) for sample collection, and the PerkinElmer (PE) workflow for nucleic acid extraction and RT-qPCR (Table S1) (7). Adapting surface sampling to this widely used clinical workflow would enable its application to an entirely new sampling modality and allow surface sampling to be incorporated into a wide range of programs in schools, prisons, nursing homes, and other high-risk, indoor settings.

## RESULTS

To assess whether conclusions drawn from our established Thermo pipeline (6) could be generalized to the more widespread PerkinElmer pipeline, we first compared the performance of both methods using contrived samples. Briefly, the Thermo pipeline collects surface swabs into Matrix tubes containing 0.5% wt/vol SDS in water, extracts nucleic acids using the Omega MagBind Viral RNA/DNA kit (SKU: M6246-03) using the KingFisher Flex platform, and detects SARS-CoV-2 presence through a miniaturized 3 $\mu$L-reaction version of the TaqPath COVID-19 Combo kit (ThermoFisher Scientific, A47814) on a QuantStudio 7 Pro qPCR machine (Thermo Fisher Scientific). The PerkinElmer pipeline follows the Emergency Use Authorization of sample collection into Viral Transport Media, requires a heat-inactivation step (65C for 15 min), extracts nucleic acids using the chemagic 360 platform and reagents, and detects SARS-CoV-2 presence using the standard 15 $\mu$L-reaction protocol of the PerkinElmer New Coronavirus Nucleic Acid Detection kit on an Analytik Jena qTOWER³ 84 G real-time PCR system.

To manufacture the contrived samples, we deposited 10 $\mu$L of a heat-inactivated SARS-CoV-2 dilution series (strain WA-1, SA-WA1/2020) on laminated cards, making

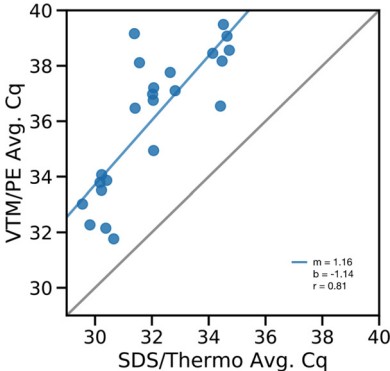

**FIG 1** Comparison of SDS/Thermo and VTM/PE pipelines on contrived samples. Average Cq values of contrived samples with the SDS/Thermo pipeline versus the Average Cq of matched samples processed through VTM/PerkinElmer pipeline. A linear regression was overlaid on the measured data (in blue) (Pearson correlation, m = 1.16, b = −1.14, r = 0.81, P = 2.87 × 10$^{-6}$). The gray line represents the expected Cq values where x = y, i.e., if the two assays performed identically on the same samples.

triplicate cards for each of the three concentrations used (206, 1024, 4096 genomic equivalents (GEs)/$\mu$L). Nine replicates were swabbed with 0.5% SDS and processed by the Thermo pipeline while the other nine were swabbed with VTM and processed by the PerkinElmer pipeline. We found that the two platforms yielded highly correlated results (Pearson correlation, r = 0.810, P = 2.87 × 10$^{-6}$) but the SDS/Thermo pipeline was more sensitive by ~4 Cq units on average (Fig. 1).

To test whether these conclusions extended to a real-world setting, we collected duplicate biological replicates of 30 samples from isolation housing in which known COVID-19 patients, confirmed by positive anterior nares RT-qPCR, were housed (IRB-approved research under HRPP UCSD protocol 200477). The distribution of sampled surfaces is given in Table 1, and their spatial localization in one of the apartments is given in Fig. 2 as an example. In a crossover protocol to separate the effects of the swabbing/transport medium and RNA extraction from the effects of the qPCR assay, we ran each protocol through RNA extraction, exchanged RNA between the UCSD lab and the SDPHL, then subjected the resulting RNA to RT-qPCR on the other platform (Table S2). This created four sets of samples summarized in Table S3.

We found that each laboratory performed best when sample extraction and RT-qPCR processing occurred in the same facility, presumably because of RNA degradation during

**TABLE 1** Number of detection events per feature per apartment

| Room Type | Feature | Apt A | Apt B | Apt C |
|---|---|---|---|---|
| Bathroom | bathroom sink | porcelain | porcelain | porcelain |
|  | bathroom door handle |  | stainless steel |  |
|  | toilet floor | ceramic | ceramic | ceramic |
| Bedroom | bed headboard | sealed wood | raw wood | sealed wood |
|  | bedside floor | carpet | vinyl | carpet |
|  | bedside table | sealed wood | wicker | sealed wood |
| Kitchen | kitchen counter |  | laminate | laminate |
|  | kitchen sink handles | stainless steel | stainless steel | stainless steel |
|  | kitchen table | laminate |  | laminate |
| Living Room | desk top | sealed wood |  | sealed wood |
|  | coffee table |  | sealed wood |  |
|  | desk chair | fabric |  |  |
|  | light switch | plastic | plastic | plastic |

4 3 2 1 0

The text within each cell indicates the material of the sampled feature and the heatmap coloring represents the counts of positive detection events from the different combinations of extraction and RT-qPCR facility. A value of 4 indicates detection in all 4 pipeline permutations, whereas 0 indicates no detection in any of the combinations.

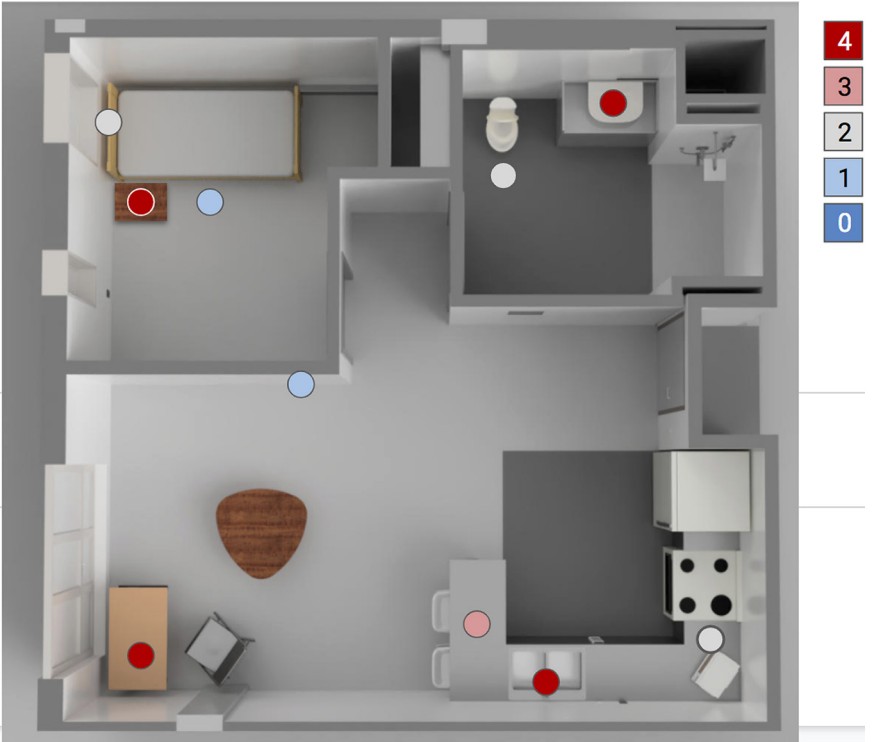

**FIG 2** Ili mapping of positive detection events across pipeline combinations on a representative 3D render of the rooms swabbed. This 3D rendering represents the relationship between rooms and features that were swabbed in Apt C (Table 1). The color scale represents the number of positive detection events returned across the combinations of extraction and RT-qPCR facilities.

transit (Fig. 3A and B). A Kruskal-Wallis H test confirmed that the mean Cq's differ significantly across the pipeline combinations (H = 26.9, $P = 6.22 \times 10^{-6}$) (Fig. 3C). Pairwise Mann-Whitney U tests between the groups showed that the Cq ranks are significantly different between groups processed at the different PCR facilities, but there is no significant difference for groups within PCR facility. The SDPHL RT-qPCR assay performed better on the samples processed with the PE pipeline than on the Thermo/PE paired samples (mean Cq difference 0.88, $U = 0.693$, $P > 0.1$), whereas the UCSD RT-qPCR assay performed better with the samples processed with the Thermo pipeline than with the paired PE/Thermo samples (mean Cq difference 1.2, $U = 1.48$, $P > 0.1$) (Fig. 3C). However, the Thermo workflow provided an advantage in sensitivity of 5.1 Cq units on average over the PE workflow yielding a $P$-value of $5.86 \times 10^{-3}$ after correcting for multiple comparisons (FDR-Benjamin/Hochberg) ($U = 3.72$). None of the assays were perfect: the Thermo pipeline detected 6 samples that the PE pipeline counted as negative, and the PE pipeline detected 2 samples that the Thermo pipeline counted as negative (Table S2). To calculate the sensitivity of each pipeline, we assumed that if at least one of the pipeline combinations (Thermo, Thermo/PE, PE, PE/Thermo) detected the presence of virus, then that sample could be considered a true positive resulting in a true positivity rate of 91% for the Thermo pipeline and 73% for the PE pipeline. However, we were unable to make any assumptions about the true negative rate and so are unable to calculate the specificity of each pipeline.

## DISCUSSION

We note that although the PE assay is less sensitive, this level of accuracy is sufficient for projects such as SASEA where the goal is to screen environments for further resource allocation for COVID-19 mitigation efforts rather than to perform a diagnostic test. In conclusion, although the optimized Thermo protocol we developed for SASEA

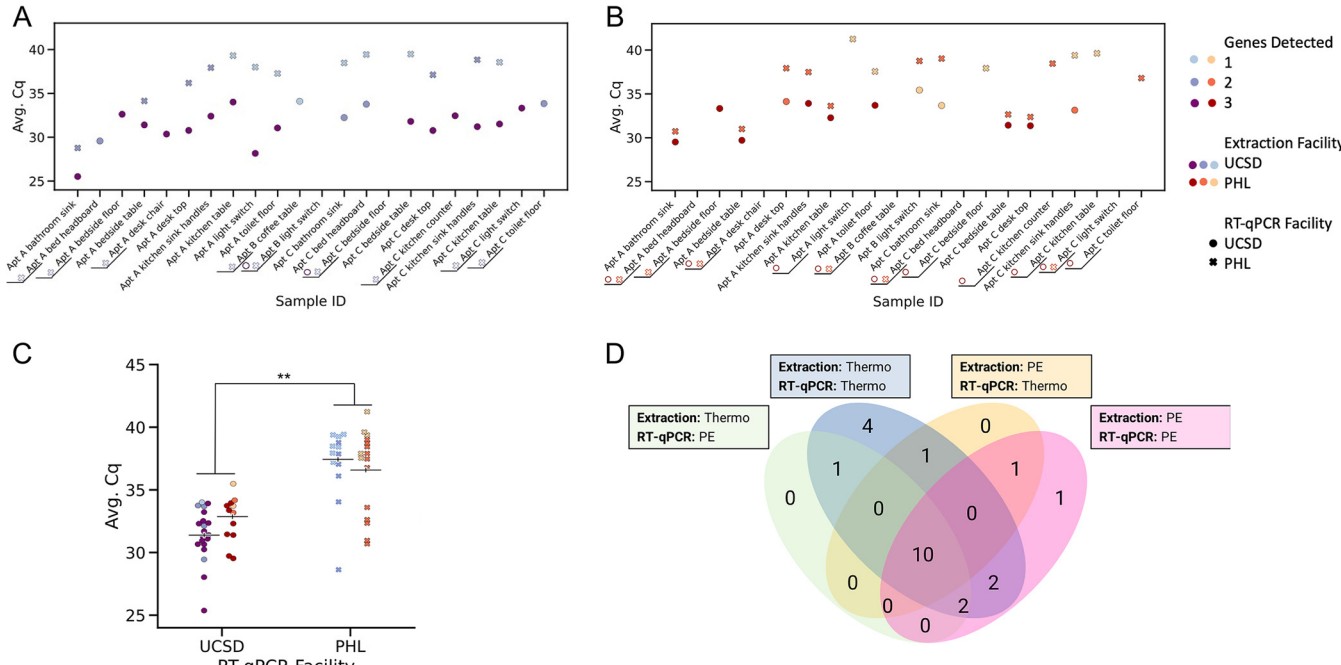

**FIG 3** Comparison of SDS/Thermo and VTM/PE pipeline combinations on real samples. (A and B) Scatterplots showing the performance of the Thermo (UCSD) and PerkinElmer (PHL) RT-qPCR workflows on surface samples extracted at both facilities. Empty x's and o's next to the sample name indicates that no viral signal was detected in that sample for that combination of extraction and RT-qPCR facility. (C) Swarm plots showing that the sensitivity of the Thermo RT-qPCR workflow is higher than that of the PE pipeline (Kruskal-Wallis, $P < 0.01$). Post hoc analysis showed that there was no significant difference between samples that underwent RT-qPCR at the same facility ($P > 0.1$) but there were differences between RT-qPCR facilities ($P < 0.05$). (D) Venn diagram showing the number of positive samples detected by each of the extraction facility/RT-qPCR pipeline combinations.

offers considerable sensitivity advantages, the PE assay is still pragmatically useful for classroom SARS-CoV-2 surveillance and can leverage large existing investments in infrastructure and expertise.

## MATERIALS AND METHODS

Surface sampling with two transport media (VTM and SDS) and subsequent nucleic extraction and SARS-CoV-2 readout with two RT-qPCR pipelines (Thermo and PerkinElmer) are compared in a factorial design.

**UCSD protocol. Nucleic acid extraction.** Individual 96-well tube racks were vortexed for 5 min at 3200 RPM to promote the suspension of viral particles from the swabs into the 0.5% wt/vol SDS solution. Afterwards, 150 $\mu$L of the suspension buffer (0.5% SDS) were transferred with a multichannel pipette into barcoded deep well extraction plates (ThermoFisher Scientific, 95040450) and processed using the Omega MagBind Viral DNA/RNA kit (Omega Bio-Tek, M6246) on the Kingfisher Flex (ThermoFisher Scientific) platform following manufacturer's protocol with the following modifications: only 150 $\mu$L of sample input was used (instead of the recommended 200 $\mu$L) and 10 $\mu$L of MS2 phage was added to each well as an extraction control.

**RT-qPCR (Multiplexed TaqPath).** Viral gene detection assays were performed using the RT-qPCR-based TaqPath COVID-19 Combo kit (ThermoFisher Scientific, A47814) on a QuantStudio 7 Pro with a 384-well sample block (ThermoFisher Scientific, A43185) according to the manufacturer's protocol with the following modifications: 2 $\mu$L of purified RNA was added to a 1 $\mu$L reaction mix containing 0.75 $\mu$L TaqPath 4× Enzyme mix (ThermoFisher Scientific, A28523), 0.15 $\mu$L multiplex probe mix, and 0.1 $\mu$L nuclease free water, for a total reaction volume of 3 $\mu$L. Low volume transfers (<5 $\mu$L) were done with Mosquito HV Liquid Handlers (SPT Labtech). The following RT-qPCR cycling conditions were used: 25°C for 2 min, 53°C for 10 min, 95°C for 2 min, 55 cycles of 95°C for 3 s, and 60°C for 30 s. The signal was measured at the end of each 30 s interval at 60°C. Baseline determination and quantification cycle (Cq) signal determination were made using the Design and Analysis v2.4.3 software (Applied Biosystems) using the relative threshold (Crt) method. Positive calls for individual gene reporters were made according to Table S3.

**SDPHL protocol. Nucleic acid extraction.** Environmental samples in VTM were heat-inactivated in a bead-bath at 65C for 15 min, then cooled at 2 to 8C for a minimum of 10 min prior to processing. Assay controls (positive, negative, and internal controls) were thawed prior to use. Additional reagents (Poly A RNA and Proteinase K) were prepared per kit instructions prior to use. Specimens were brought to room temperature, placed into sample racks, and loaded onto the Janus Reformatter Robot. From the Janus Reformatter, samples and their corresponding reagent plates were transferred to the Chemagic 360 instrument for extraction by the manufacturer's protocol.

**RT-qPCR.** Extract plates from the Chemagic 360 were transferred to the Janus qPCR Workstation Robot, along with qPCR master mix reagents, and loaded into a 384-qPCR plate. Viral gene detection assays were performed using the RT-qPCR-based PerkinElmer New Coronavirus Nucleic Acid Detection kit (2019-nCOV-PCR-AUS) on an Analytik Jena qTOWER³ 84 G real-time PCR system with a 384-well sample block according to the manufacturer's protocol for 15 $\mu$L reactions. The following RT-qPCR cycling conditions were used: 37C for 2 min, 50C for 5 min, 42C for 35 min, 94C for 10 min, 45 cycles of 94C for 10 s, 55C for 15 s, 65C for 45 s. The .trf method file generated by the Janus qPCR Workstation Robot was copied and transferred to the Analytic Jena as the qPCR method. Results were analyzed following assay processing. Interpretation of results was performed using qPCRSoft 384 for the Analytik Jena.

Fluorophore probes used for detection of two COVID targets (N [nucleocapsid] gene and ORF1ab [open reading frame 1 ab]) were FAM and Rox, respectively. The IC (bacteriophage MS2) used a Hex fluorophore probe. Thresholds used for interpretation were between 5 and 15 dRn (delta in normalized reporting value). If a sample had either COVID target detected with Ct values below 42 at a threshold of 15, the sample was determined positive. Samples with detectable Internal Control but without detectable values for either of the two COVID targets were reported as negative. Samples which only had COVID targets within 'detectable' range at a threshold of 5 but outside 'detectable' range at threshold of 15 (Ct above 42) were considered inconclusive; these were treated as 'negative' for reporting. Samples without positive values for either of the two COVID targets in addition to failed Internal Controls were considered invalid and reported as such.

## SUPPLEMENTAL MATERIAL

Supplemental material is available online only.

**TABLE S1**, DOCX file, 0.01 MB.
**TABLE S2**, DOCX file, 0.02 MB.
**TABLE S3**, DOCX file, 0.01 MB.

## ACKNOWLEDGMENTS

This research was supported by NIH grant (K01MH112436) to RFM, the County of San Diego Health and Human Services Agency (Contract 563236), and the Career Award for Medical Scientists from the Burroughs Wellcome Fund to A.F.C. We thank Marisol Chacon, Sydney C. Morgan, Alhakam Nouri, Ashley Plascencia, Christopher A. Ruiz, and Lizbeth Franco Vargas for their support with environmental SARS-CoV-2 detection as part of the EXCITE Lab.

The following reagent was deposited by the Centers for Disease Control and Prevention and obtained through BEI Resources, NIAID, NIH: SARS-Related Coronavirus 2, Isolate USA-WA1/2020, NR-52281.

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
