## [Reviewer comments · mSystems]

Implementation of Practical Surface SARS-CoV-2 Surveillance in School Settings

Victor Cantú, Pedro Belda-Ferre, Rodolfo Salido, Rebecca Tsai, Brett Austin, William Jordan, Menka Asudani, Amanda Walster, Celestine Magallanes, Holly Valentine, Araz Majnoonian, Carrissa Wijaya, Vinton Omaleki, Karenina Sanders, Stefan Aigner, Nathan Baer, Maryan Betty, Anelizze Castro-Martínez, Willi Cheung, Evelyn Crescini, Peter De Hoff, Emily Eisner, Abbas Hakim, Bhavika Kapadia, Alma Lastrella, Elijah Lawrence, Toan Ngo, R. Ostrander, Shashank Sathe, Phoebe Seaver, Elizabeth Smoot, Aaron Carlin, Gene Yeo, Louise Laurent, Anna Liza Manlutac, Rebecca Fielding-Miller, and Rob Knight

Corresponding Author(s): Rob Knight, UCSD School of Medicine

Review Timeline:

Submission Date:	January 28, 2022
Editorial Decision:	April 20, 2022
Revision Received:	April 26, 2022
Accepted:	May 2, 2022

Editor: Ileana Cristea

Reviewer(s): Disclosure of reviewer identity is with reference to reviewer comments included in decision letter(s). The following individuals involved in review of your submission have agreed to reveal their identity: Jiaxian Shen (Reviewer #1)

Transaction Report:

DOI: <https://doi.org/10.1128/mSystems.00103-22>

April 20, 2022

Prof. Rob Knight
UCSD School of Medicine
9500 Gilman Drive
MC 0602
La Jolla, CA 92093

Re: mSystems00103-22 (Implementation of Practical Surface SARS-CoV-2 Surveillance in School Settings)

Dear Prof. Rob Knight:

Thank you for submitting your manuscript to mSystems. We have completed our review and I am pleased to inform you that, in principle, we expect to accept it for publication in mSystems. Although we have invited ten potential reviewers, we have been able to secure only one review. However, as you will see, this review is thorough and provides critical feedback on the manuscript. We invite the submission of a revised manuscript that fully addresses the reviewer's comments.

Preparing Revision Guidelines

Sincerely,

Ileana Cristea

Editor, mSystems

Journals Department
Reviewer comments:

This study compared two workflows (SDS/Thermo and VTM/PE) for surface surveillance of SARS-CoV-2 virus, using both contrived and realistic samples. This study is valuable to combat pandemic spread and keep people in indoor public settings safe. While the study is beneficial, it lacks important explanations so that the research outcomes can be readily leveraged.

For example, the manuscript claimed that the authors' method shows superior sensitivity and specificity than the SDPHL pipeline. While I can tell from the manuscript that the SDS/Thermo pipeline is more sensitive, I do not see the authors discuss and compare the specificity between the two pipelines. Adding contents about specificity may be necessary.

Discussions on testing using realistic samples are generally solid. However, the strength of the arguments needs to be improved. 1) More explicit references to figures are needed to better demonstrate the connection between evidence and interpretation. 2) Elaborations are needed for a few arguments. Specifically,

L147-156 in general: Please refer to the specific panel in figure 3 to back up the interpretations.

L147-148: Specify the conclusion that each laboratory performed best when sample extraction and RT-qPCR processing occurred in the same facility is in terms of the number of positive samples (Fig. 3D), average Cq (Fig. 3A&B?), or something else.

L149-152: 1) If I understand it correctly, this was interpreted from Fig. 3C. If so, clearly refer to Fig. 3C so that no one will be confused. 2) It will be better to mark out where the mean Cq is for each group in Fig. 3C (maybe using a short horizontal line). Right now, it is unclear whether the SDPHL RT-qPCR assay performed better on the VTM swabs or on the SDS swabs by just looking at the figure. 3) Both Cq differences are not statistically significant. Why is 1.2 substantially better while 0.88 only slightly better?

L152-154 (However...Fig. 3): 1) What is the comparison group that leads to the conclusion of the substantial advantage in sensitivity of 5.1 Cq units? VTM/PE combination? 2) Please elaborate on the statistical testing done here and in Fig. 3C. Are the two (Kruskal-Wallis, $p < 0.01$) results of the same testing? In L152-154, if the conclusion is drawn by comparing SDS/Thermo with VTM/PE, post-hoc tests should be performed following Kruskal-Wallis. In Fig. 3C, are samples extracted in both UCSD and PHL treated as the same group? In other words, is the testing performed based on four groups or two? If the former, post-hoc tests should be conducted as well. And the significance representation may need to be modified. If the latter, nonparametric equivalent of t test is more appropriate, e.g., Wilcoxon Test.

Figure 3C: The significance representation is confusing. By convention, one asterisk (*) means $0.01 < p \leq 0.05$, but the caption clearly mentioned $p < 0.01$.

L160-162: Please explain why sampling twice can compensate for the sensitivity gap between VTM/PE assay and SDS/Thermo assay.

Additionally, since this comparison study targets practical implementation of the pipelines, I have the following suggestions to improve its clarity.

Uniform the wording

After reading through the manuscript, I understand that there are two workflows and in the crossover comparison, RT-qPCR assays were exchanged while swabbing/transport medium and RNA extraction remain unchanged.

Lab	Sample collection	RNA extraction	RT-qPCR
UCSD	SDS	Thermo pipeline	Thermo pipeline
SDPHL	VTM	Perkin-Elmer pipeline	Perkin-Elmer pipeline

This is clear when it was first introduced in L141-145. But downstream, the inconsistency of wording adds confusion. I suggest defining the wording first and sticking to the defined terms in the rest of the manuscript. For example, as per the current manuscript, confusion might occur in the following instances.

- In L149-150, the term “VTM swabs” actually represent samples collected in VTM and extracted following the Perkin-Elmer pipeline, while according to L114-125, RNA extraction step reads like part of the processing pipeline rather than falling under the name of swabbing/transport medium. Similarly, do the VTM and SDS in Supplementary Table S1 also represent from sample collection to RNA extraction?
- In Figure 3 and Supplementary Table S1, lab names/locations (UCSD, PHL) were used to represent the pipeline while in Results, the pipeline names (SDS/Thermo, VTM/PE) and lab names seemed to be used interchangeably. Furthermore, wording of lab names was not standardized, e.g., PHL in Figure 3 and County in Supplementary Table S1.

Other comments:

L307: Recommend using $p = 2.87 \times 10^{-6}$ rather than $p < 0.01$ in the figure caption to match what is used in L132-133.

Table 1 & Figure 2: The idea of using a heatmap to represent the counts of positive detection events is great. But the color differences among 3, 2, and 1 are not large

enough for easy distinguishment. Try to make it clearer. Maybe switching to a 2-color scale would help?

Figure 3: The sequential color scheme to indicate the number of gene targets is hard to distinguish, especially for PHL.

Figure 3 A&B: 1) It would be more visually friendly to order the x axis consistently across A and B for sampling locations (i.e., sample names). 2) I suggest including the sample names in x axis for those with no detection and using some type of label to indicate that they are not detected so that the readers are better informed.

For protocol clarity, please include the following information in the UCSD protocol.

1. L179: specify the vortex speed
2. L180: include w/v as well when mentioning 0.5% SDS here.
3. L182 & L192: provide cat# of barcoded deep well extraction plates & 384-well sample block

UNIVERSITY of CALIFORNIA, SAN DIEGO
SCHOOL OF MEDICINE

Rob Knight, Ph.D.
Professor
Department of Pediatrics
UC San Diego School of Medicine

9500 Gilman Drive MC0763
La Jolla, California 92093
Tel: (619) 543-7900
E-mail: robknight@ucsd.edu

April 26, 2022

Dear Ileana,

Thank you very much for overseeing the review of our manuscript. We have carefully read all of the reviewers' critiques and revised our manuscript in response to their comments (responses shown in blue, manuscript changes tracked in 'Marked-Up Manuscript', and line references point to the revised manuscript). We believe the manuscript has been significantly strengthened due to these changes and is now suitable for publication in *mSystems*. We have included below a point-by-point response to the reviewer's comments.

Sincerely,

Rob Knight

Rob Knight, Ph.D.
Director, Center for Microbiome Innovation
Professor, Department of Pediatrics, Bioengineering, and Computer Science and Engineering UC San Diego

We thank the reviewer for their time and for carefully reviewing our manuscript. We believe their comments have led to us seeking out references that greatly improved the strength of the paper. We have taken their constructive comments and edited our manuscript appropriately.

Summary/Overview:

This study compared two workflows (SDS/Thermo and VTM/PE) for surface surveillance of SARS-CoV-2 virus, using both contrived and realistic samples. This study is valuable to combat pandemic spread and keep people in indoor public settings safe. While the study is beneficial, it lacks important explanations so that the research outcomes can be readily leveraged.

For example, the manuscript claimed that the authors' method shows superior sensitivity and specificity than the SDPHL pipeline. While I can tell from the manuscript that the SDS/Thermo pipeline is more sensitive, I do not see the authors discuss and compare the specificity between the two pipelines. Adding contents about specificity may be necessary.

We appreciate this suggestion and clarified the sensitivity of our pipeline on lines 162-168.

Discussions on testing using realistic samples are generally solid. However, the strength of the arguments needs to be improved. 1) More explicit references to figures are needed to better demonstrate the connection between evidence and interpretation. 2) Elaborations are needed for a few arguments. Specifically,

L147-156 in general: Please refer to the specific panel in figure 3 to back up the interpretations.

Thank you for this suggestion. We have referenced the specific panels in Figure 3 (A and B), which provide support for our interpretations, as suggested (line 148).

L147-148: Specify the conclusion that each laboratory performed best when sample extraction and RT-qPCR processing occurred in the same facility is in terms of the number of positive samples (Fig. 3D), average Cq (Fig. 3A&B?), or something else.

We appreciate this suggestion, and we have incorporated the reference to Figure 3C (line 149), which this statement is based on (although the other panels mentioned also support it).

L149-152: 1) If I understand it correctly, this was interpreted from Fig. 3C. If so, clearly refer to Fig. 3C so that no one will be confused.

We appreciate this feedback and have incorporated the reference to Figure 3C as suggested (line 149).

2) It will be better to mark out where the mean Cq is for each group in Fig. 3C (maybe using a short horizontal line). Right now, it is unclear whether the SDPHL RT-qPCR assay performed better on the VTM swabs or on the SDS swabs by just looking at the figure. 3) Both Cq differences are not statistically significant. Why is 1.2 substantially better while 0.88 only slightly better?

We have added a short horizontal line to each group in Fig. 3C as suggested. We were informally using the threshold of 1 Cq unit for "substantially" versus "slightly", but agree that this is not well justified, so we have deleted "substantially" and "slightly" and simply report them as better (lines 152-156).

L152-154 (However...Fig. 3): 1) What is the comparison group that leads to the conclusion of the substantial advantage in sensitivity of 5.1 Cq units? VTM/PE combination? 2) Please elaborate on the statistical testing done here and in Fig. 3C. Are the two (Kruskal-Wallis, $p < 0.01$) results of the same testing? In L152-154, if the conclusion is drawn by comparing

SDS/Thermo with VTM/PE, post-hoc tests should be performed following Kruskal-Wallis. In Fig. 3C, are samples extracted in both UCSD and PHL treated as the same group? In other words, is the testing performed based on four groups or two? If the former, post-hoc tests should be conducted as well. And the significance representation may need to be modified. If the latter, nonparametric equivalent of t test is more appropriate, e.g., Wilcoxon Test.

We appreciate the reviewer’s detailed attention to the statistical analysis. We have now clarified that the comparison group that leads to the advantage of 5.1 Cq units is indeed drawn by comparing the Thermo and PE workflows as the reviewer appreciates (lines 144-146). We have added a note to clarify this. The Kruskal-Wallis test was performed across all four combinations and was found to be significant. Performing a post-hoc test comparing each combination of groups directly, via Wilcoxon Rank-Sum, yields a P-value of 5.86×10^{-3} for the Thermo vs PE pipelines and we have added this to the manuscript on line 158. We have also added in the Wilcoxon Rank-Sum results for the other combinations on lines 148-159. We have clarified our statistical analysis in the caption of Figure 3C.

Results of Wilcoxon Rank-Sum pairwise

Figure 3C: The significance representation is confusing. By convention, one asterisk (*) means $0.01 < p \leq 0.05$, but the caption clearly mentioned $p < 0.01$

We apologize for the error and have fixed the inconsistency.

L160-162: Please explain why sampling twice can compensate for the sensitivity gap between VTM/PE assay and SDS/Thermo assay.

On further analysis, we agree that this may not solve the issue, depending on whether errors that lead to dropout are stochastic or deterministic, and have deleted this suggestion accordingly.

Additionally, since this comparison study targets practical implementation of the pipelines, I have the following suggestions to improve its clarity.

Uniform the wording

After reading through the manuscript, I understand that there are two workflows and in the crossover comparison, RT-qPCR assays were exchanged while swabbing/transport medium and RNA extraction remain unchanged.

Lab	Sample collection	RNA extraction	RT-qPCR
UCSD	SDS	Thermo pipeline	Thermo pipeline
SDPHL	VTM	Perkin-Elmer pipeline	Perkin-Elmer pipeline

This is clear when it was first introduced in L141-145. But downstream, the inconsistency of wording adds confusion. I suggest defining the wording first and sticking to the defined terms in the rest of the manuscript. For example, as per the current manuscript, confusion might occur in the following instances.

- In L149-150, the term “VTM swabs” actually represent samples collected in VTM and extracted following the Perkin-Elmer pipeline, while according to L114-125, RNA extraction step reads like part of the processing pipeline rather than falling under the name of swabbing/transport medium. Similarly, do the VTM and SDS in Supplementary Table S1 also represent from sample collection to RNA extraction?
- In Figure 3 and Supplementary Table S1, lab names/locations (UCSD, PHL) were used to represent the pipeline while in Results, the pipeline names (SDS/Thermo, VTM/PE) and lab names seemed to be used interchangeably. Furthermore, wording of lab names was not standardized, e.g., PHL in Figure 3 and County in Supplementary Table S1.

We thank the reviewer for catching these inconsistencies and have corrected them as recommended.

Other comments:

L307: Recommend using $p = 2.87 \times 10^{-6}$ rather than $p < 0.01$ in the figure caption to match

what is used in L132-133.

Thank you for pointing this out. The specific p-value has been added to the figure caption.

Table 1 & Figure 2: The idea of using a heatmap to represent the counts of positive detection events is great. But the color differences among 3, 2, and 1 are not large enough for easy distinguishment. Try to make it clearer. Maybe switching to a 2-color scale would help?

We have switched to a 2-color scale as suggested.

Figure 3: The sequential color scheme to indicate the number of gene targets is hard to distinguish, especially for PHL.

We have switched to a color scheme that highlights differences between adjacent bands rather than the sequential color scheme used.

Figure 3 A&B: 1) It would be more visually friendly to order the x axis consistently across A and B for sampling locations (i.e., sample names). 2) I suggest including the sample names in x axis for those with no detection and using some type of label to indicate that they are not detected so that the readers are better informed.

We have implemented these suggestions, which we appreciate.

For protocol clarity, please include the following information in the UCSD protocol.

1. L179: specify the vortex speed
2. L180: include w/v as well when mentioning 0.5% SDS here.
3. L182 & L192: provide cat# of barcoded deep well extraction plates & 384-well sample block

We have provided all these clarifications as requested (vortex speed, line 189; cat#, line 193).

May 2, 2022

Prof. Rob Knight
UCSD School of Medicine
9500 Gilman Drive
MC 0602
La Jolla, CA 92093

Re: mSystems00103-22R1 (Implementation of Practical Surface SARS-CoV-2 Surveillance in School Settings)

Dear Prof. Rob Knight:

Congratulations! Your manuscript has been accepted, and I am forwarding it to the ASM Journals Department for publication. For your reference, ASM Journals' address is given below. Before it can be scheduled for publication, your manuscript will be checked by the mSystems production staff to make sure that all elements meet the technical requirements for publication. They will contact you if anything needs to be revised before copyediting and production can begin. Otherwise, you will be notified when your proofs are ready to be viewed.

Publication Fees:

We recognize that the video files can become quite large, and so to avoid quality loss ASM suggests sending the video file via <https://www.wetransfer.com/>. When you have a final version of the video and the still ready to share, please send it to mSystems staff at mssystems@asmusa.org.

For mSystems research articles, if you would like to submit an image for consideration as the Featured Image for an issue, please contact mSystems staff at mssystems@asmusa.org.

Sincerely,

Ileana Cristea
Editor, mSystems

Journals Department
Table S3: Accept
Table S1: Accept
Table S2: Accept